# DQG: Database Question Generation for Exact Text-based Image Retrieval

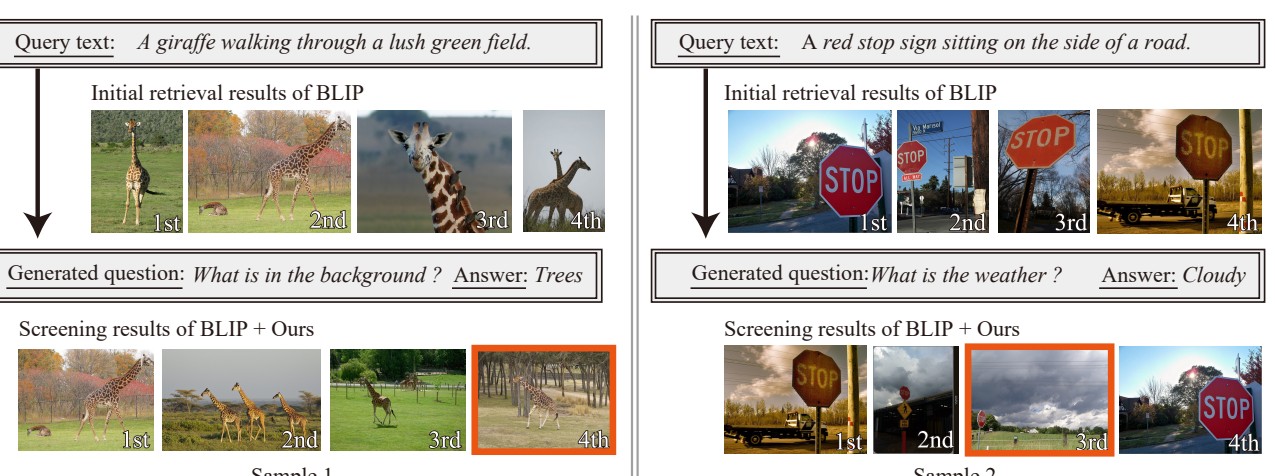

**Figure 1: Retrieval result examples of our approach. The image surrounded by the orange frame indicates the ground truth image paired with the query text. Our database question generation (DQG) approach generates questions that can screen similar but non-target images by analyzing the target DB. In the left example, initial results show diverse backgrounds, while ours consistently features tree-filled backgrounds, ranking the ground truth image 4th. In the right example, initial results show varied weather conditions, whereas ours consistently shows cloudy weather, ranking the ground truth image 3rd.**

## ABSTRACT

Screening similar but non-target images in text-based image retrieval is crucial for pinpointing the user's desired images accurately. However, conventional methods mainly focus on enhancing text-image matching performance, often failing to identify images that exactly match the retrieval intention because of the query quality. User-provided queries frequently lack adequate information for screening similar but not target images, especially when the target database (DB) contains numerous similar images. Therefore, a novel approach is needed to extract valuable information from users for effective screening. In this paper, we propose a DB question generation (DQG) model to enhance exact cross-modal image retrieval performance. Our DQG model learns to generate effective questions that precisely screen similar but non-target images using DB contents information. By answering the questions generated from our model, users can reach their desired images by only answering the presented questions even within DBs with similar content. Experimental results on publicly available datasets show that our

proposed approach can significantly improve exact cross-modal image retrieval performance. Code is available in the supplemental materials and will be publicly available.

## CCS CONCEPTS

• **Information systems** → **Image search**; **Task models**; *Query reformulation*; *Content analysis and feature selection*; Multimedia databases; • **Human-centered computing** → Interaction design theory, concepts and paradigms; • **Computing methodologies** → *Computer vision*; *Natural language generation*.

## KEYWORDS

Interactive image retrieval, image re-ranking, question answering

**ACM Reference Format:**
Anonymous Author(s). 2018. DQG: Database Question Generation for Exact Text-based Image Retrieval. In *Proceedings of MM 23Proceedings of the 32nd ACM International Conference on Multimedia (MM'24), October 28-November 1, 2024, Melbourne, Australia.* ACM, New York, NY, USA, 10 pages. https://doi.org/XXXXXXX.XXXXXXX

## 1 INTRODUCTION

Image retrieval through text query, also known as text-based image retrieval, is one of the most fundamental research topics that should grasp the user's retrieval purpose [30, 58]. The primal aim is to retrieve images similar to the provided query, commonly known as similar image retrieval, and it is a critical task for various Web

services such as search engines, recommendation systems, and e-commerce services [7, 13, 60]. Recent advancements in text-based image retrieval have primarily focused on enhancing the performance of similar image retrieval by leveraging shared latent spaces to compute similarities between query texts and candidate images [43]. While these methods excel in similar image retrieval, they often fall short in facilitating the retrieval of specific target images that precisely match users' retrieval intentions, referred to as exact image retrieval. This challenge arises due to the difficulty users face in providing sufficiently detailed query information.

Exact image retrieval works well when users provide an appropriate query that contains ample information to uniquely identify the desired image from the target database (DB). However, consistently providing such precise queries poses challenges, given users' limited knowledge about the entire target DB's content. User-provided queries can correspond to multiple candidate images, leading to decrease retrieval performance, even in state-of-the-art text-image foundation models like CLIP [33] and BLIP [24]. This problem escalates significantly when the target DB contains abundant similar content, as it increases the number of candidate images related to queries. For instance, retrieving a specific desired image using the text query "A photo of travel with my friends" from a DB abundant in travel images poses significant challenges. Given the diverse applications of exact image retrieval such as lifelog search [46] and landmark search [53], dealing with the appropriate query provision difficulty is an essential topic.

One promising approach to alleviate query provision challenges is integrating a question-generation (QG) scheme into conventional text-based image retrieval methods. In the adjacent field of document information retrieval, the QG-based retrieval scheme has been widely discussed for enhancing exact information retrieval performance in addition to the conventional retrieval scheme [1, 56]. By introducing a QG module, retrieval systems can better understand users' retrieval intentions through text-based interactions, enabling users to provide additional clues for finding specific target information by answering provided questions. Referring to the performance improvement in the document information retrieval field, the QG module is likely to contribute to exact image retrieval performance improvement by alleviating the query provision difficulty.

The QG module for exact image retrieval needs to adaptively generate questions tailored to each target DB. When generated questions align with multiple candidate images, similar but non-target images cannot be screened even though the retrieval system receives the answers from users. Therefore, the QG module for exact image retrieval must generate questions capable of screening similar candidate images within the target DB adaptively. While manually preparing predefined question labels is a theoretical solution for adaptive QG, it demands considerable effort due to the diverse nature of image retrieval across different target DBs. Consequently, an adaptive QG module that learns suitable questions solely from target DB contents is highly desirable.

In this paper, we propose a DB question generation (DQG) approach for enhancing exact text-based image retrieval. Our approach focuses on learning the appropriate question on the target DB only from those candidate images. Here, the appropriate questions must possess two key characteristics: effectively screening similar candidate images and being grammatically understandable to users. Our DQG module generates questions from DB information input, and the generated questions are optimized to meet the two characteristics by minimizing retrieval-based loss and grammatical-based loss. Specifically, our learning scheme transfers the knowledge of the other vision and language models solely using the candidate images on DBs, enabling automatic question generation without retrieval task-specific question labels. The generated questions by our DQG module guide users to their desired images by screening similar but non-target candidate images based on user responses. In other words, users can reach their desired images by only answering the presented questions with our approach. Experimental results show that our approach can improve the exact image retrieval performance and generate grammatically correct questions.

The contributions of this paper are summarized as follows.

**DQG approach for exact text-based image retrieval**
Our DQG approach effectively screen similar but non-target candidate images, guiding users to their desired images through provided questions.

**DQG learning scheme without task-specific labels**
Our DQG module is trained to generate the suitable questions from DB information input without using any question labels for retrieval tasks.

**Significant performance improvement**
Experimental results demonstrate notable enhancement in exact image retrieval performance compared to the current state-of-the-art BLIP method, by +13.1%, +7.5%, and +10.4% in the R@1 metric on the MSCOCO, VG, and biased-MSCOCO datasets, respectively.

## 2 RELATED WORKS

### 2.1 Text-based image retrieval

Our approach focuses on screening similar but non-target candidate images in a text-based image retrieval (hereinafter referred to as TBIR) paradigm. Most recent TBIR methods aim to map a query text and candidate images into the latent space $\mathcal{E}$ [5, 11, 37, 42]. That is, they train the two mapping functions $\text{Enc}^{\mathcal{L}}(\cdot): \mathcal{L} \rightarrow \mathcal{E}$ and $\text{Enc}^{\mathcal{V}}(\cdot): \mathcal{V} \rightarrow \mathcal{E}$, where $\mathcal{L}$ and $\mathcal{V}$ are lingual and visual spaces, respectively. By embedding a query text and candidate images via the trained mapping functions $\text{Enc}^{\mathcal{L}}(\cdot)$ and $\text{Enc}^{\mathcal{V}}(\cdot)$, similarities between the text query and candidate images are calculated on the space $\mathcal{E}$.

Conventionally, TBIR is realized based on statistical correlation analysis, such as canonical correlation analysis [18]. However, the advent of deep neural network technologies has revolutionized representation learning, enabling the training of robust mapping functions $\text{Enc}^{\mathcal{L}}(\cdot)$ and $\text{Enc}^{\mathcal{V}}(\cdot)$ [12, 20, 22, 24, 33, 59]. Based on the hinge loss, Kiros *et al.* [22] trained $\text{Enc}^{\mathcal{L}}(\cdot)$ and $\text{Enc}^{\mathcal{V}}(\cdot)$ so that the similarities between correct text-image pairs are higher than those between other different pairs. Faghri *et al.* [12] improved the method used by Kiros *et al.* [22] by considering the number of images between a text query and the retrieval target image. The existing TBIR performance has been improved by focusing on the loss function and network architectures.

While existing TBIR methods excel in similar image retrieval tasks, our work targets the specific retrieval of images exactly matching users' intentions. By integrating DQG-based interaction for re-ranking, our system leverages user feedback to screen similar but non-target images more effectively.

## 2.2 Re-ranking

Re-ranking has been studied in various retrieval tasks such as person re-identification and object retrieval, and methods for TBIR tasks have also been proposed [3, 4, 30, 36, 54]. Our retrieval approach aligns with re-ranking methodologies as it incorporates additional user information for screening similar candidate images. Therefore, we review the difference between our approach and similar works in this subsection. Based on the necessity of user interaction, re-ranking methods can be roughly classified into two categories: self re-ranking and feedback-based re-ranking.

Self re-ranking aims to improve retrieval performance by estimating the key information from the top-ranked images of the initial retrieval results. Several self-re-ranking methods [44, 45, 55] rank text labels in reverse order utilizing each initial ranked image as a query and re-rank the initial retrieval results based on these results. Although these self-re-ranking methods can improve the initial retrieval performance without any feedback information, they cannot obtain additional information from users. Therefore, it is difficult for self-re-ranking methods to deal with query provision difficulty.

Feedback-based re-ranking aims to improve retrieval performance based on user feedback. Several learning-based methods [17, 39, 41, 50–52] allow users to provide feedback on retrieval results via natural language. In the fashion domain, Guo *et al.* proposed a reinforcement learning-based re-ranking method [17]. By learning the texts that describe differences of images, they can estimate user desired images from feedback comments on the top-ranked image. Although its method enables users to provide natural language-based feedback, there is no guarantee that the feedback provided by the users effectively clarifies their query text. Besides, users are required to consider additional natural language-based queries for the re-ranking. Also, as the most relevant methods, Yanagi *et al.* [50, 52] proposed re-ranking methods that receive information only about objects in the target image. Although these methods receive additional information for screening similar candidate images, their retrieval performance heavily depends on the objects in the database, and then the feedback is not always effective.

Following the growth of the document information retrieval field [1, 56], we introduce the DQG for exact TBIR settings. Our approach estimates the most effective questions by learning the target DB without using any labels for the retrieval task. With this procedure, users can reach their desired single image effectively by simply answering the presented questions.

## 3 DQG FOR EXACT TEXT-BASED IMAGE RETREIVAL

Our approach consists of the following two steps: initial text-based image retrieval and DQG-based screening. An overview of our approach is shown in Fig. 2. By transferring the knowledge of the other vision and language tasks, our approach learns how to generate adequate questions for the screening in the target DB without using any labels for retrieval. First, we calculate the initial retrieval

results by computing the similarities between the query text T and candidate images $I_n$ ($n = 1, \ldots, N$; $N$ being the number of candidate images). Based on the initial retrieval results, we calculate the features of the target DB using the DQG encoder module $\mathcal{G}^{\text{en}}(\cdot)$. Then, the calculated DB features are passed into the DQG decoder module $\mathcal{G}^{\text{dec}}(\cdot)$, and the $\mathcal{G}^{\text{dec}}(\cdot)$ generates free-form questions for the screening. Next, we receive feedback answers toward the generated question. Here, in the training phase, it takes a lot of costs to manually prepare feedback answers until the training convergence, we obtain the feedback answer from the QA modules by assuming them as users. Finally, the retrieval results are calculated based on the feedback answer.

## 3.1 The pre-training on pretext tasks

To learn how to generate adequate questions without using question labels for retrieval, we transfer the knowledge of various vision and language modules pre-trained on pretext tasks. As a preparation, we pre-train the image description module I2T($\cdot$), the DQG encoder and decoder modules $\{\mathcal{G}^{\text{en}}(\cdot), \mathcal{G}^{\text{dec}}(\cdot)\}$, and the QA word encoder and decoder modules $\{\mathcal{A}^{\text{word}}(\cdot), \mathcal{A}^{\text{dec}}(\cdot)\}$ with various vision and language tasks. The image description module I2T($\cdot$) is pre-trained on the image captioning pretext task [49] using the text-image paired dataset [28]. Also, the DQG and QA modules are pre-trained on the visual question generation and answering pretext tasks [2, 26] using the QA-image paired dataset [16]. Here, the QA-image paired dataset contains $C$ dictionary words $\text{dic}_c$ (e.g., "car" and "dog"), and the pre-trained $\mathcal{A}^{\text{word}}(\cdot)$ can convert these words into $D^{\text{word}}$-dimensional features $f_c^{\text{word}}$ as follows:

$$F^{\text{word}} = [f_1^{\text{word}}, \ldots, f_C^{\text{word}}]^\top, \qquad (1)$$

$$f_c^{\text{word}} = \mathcal{A}^{\text{word}}(\text{dic}_c). \qquad (2)$$

The calculated $F^{\text{word}}$ is utilized for connecting the question generation and answering modules in the following subsections.

We focus on training the DQG decoder module $\mathcal{G}^{\text{dec}}(\cdot)$ to generate questions that can effectively screen similar but non-target images. Besides, the generated questions should be grammatically correct since we provide the generated questions to users. For guaranteeing the grammatical correctness of the generated questions, we introduce the discriminative module $\mathcal{D}(\cdot)$ that can classify the real and generated questions. Specifically, we train the discriminative module $\mathcal{D}(\cdot)$ using the real questions $q^{\text{real}}$ contained in the QA-image paired dataset. The trained $\mathcal{D}(\cdot)$ is used for the loss calculation. In the training, the real questions are converted to grammatically incorrect questions $q^{\text{gen}}$ via word swapping, erasing, or inserting. $\mathcal{D}(\cdot)$ is trained for minimizing the binary cross-entropy loss $L^{\text{pre}}$ as follows:

$$L^{\text{pre}} = -(\log \mathcal{D}(q^{\text{real}}) + \log(1 - \mathcal{D}(q^{\text{gen}}))). \qquad (3)$$

After the training, it is expected that $\mathcal{D}(\cdot)$ can classify the real and generated questions. In the following sections, we refer the trained discriminative module as $\hat{\mathcal{D}}(\cdot)$. The loss calculation with $\hat{\mathcal{D}}(\cdot)$ is described in Subsec. 3.3.

## 3.2 Retrieval procedure

**Initial text-based image retrieval.** We calculate the initial ranking of the candidate images $I_n$ from a query text T. Theoretically,

**Figure 2: Overview of the proposed approach. At first, we calculate the initial retrieval results by computing the similarities between the query text and candidate images. Based on the calculated initial results, we aggregate the visual features of the candidate images using the QG encoder module $\mathcal{G}^{\mathbf{en}}(\cdot)$. The aggregated visual features are passed into the QG decoder module $\mathcal{G}^{\mathbf{dec}}(\cdot)$, and the QG decoder module generates questions for the screening. Next, our approach obtains the feedback answer from the QA modules by assuming them as users. Finally, the retrieval results are calculated based on the feedback answer.**

since an arbitrary text-based image retrieval method can be used for the first step, we explain our approach with reference to the most basic text-based image retrieval architecture [22].

First, text and image features ($f^{\mathcal{L}} \in \mathcal{R}^{D_{\mathcal{E}}}$ and $f_n^{\mathcal{V}} \in \mathcal{R}^{D_{\mathcal{E}}}$) are calculated from T and $I_n$, respectively, where $D_{\mathcal{E}}$ represents the dimension of the embedded features. Using the two trained embedding functions $\text{Enc}^{\mathcal{L}}(\cdot)$ and $\text{Enc}^{\mathcal{V}}(\cdot)$, which are provided by the arbitrary conventional text-based image retrieval method, we embed T and $I_n$ into the shared latent space as follows: $f^{\mathcal{L}} = \text{Enc}^{\mathcal{L}}(\text{T}), f_n^{\mathcal{V}} = \text{Enc}^{\mathcal{V}}(\text{I}_n)$. Then, the proposed approach calculates the cosine similarities $s_n$ between embedded features $f^{\mathcal{L}}$ and $f_n^{\mathcal{V}}$. We rank the candidate images $I_n$ as $R_k$ ($k = 1, \dots, N$) in descending order of $s_n$. Namely, $R_k$ represents the $k$th ranked candidate image using T as a query. The initial retrieval results $R_k$ are provided to the users. Besides, the users can conduct the screening if they are not satisfied with the initial retrieval results $R_k$.

**DQG-based screening.** We screen the initial retrieval results $R_k$ via DQG model. The proposed approach firstly calculates DB features $f^{\mathcal{G}} \in \mathcal{R}^{D_{\mathcal{G}}}$ from $R_k$ using the DQG encoder module $\mathcal{G}^{\text{en}}(\cdot)$ as follows:

$$f^{\mathcal{G}} = \frac{1}{\sum_k \alpha^{k-1}} \sum_k \alpha^{k-1} f_k^{\mathcal{G}}, \qquad (4)$$

$$f_k^{\mathcal{G}} = \mathcal{G}^{\text{en}}(R_k), \qquad (5)$$

where $\alpha$ is a hyperparameter that balances the importance of ranking. The calculated DB features $f^{\mathcal{G}}$ are then passed into the DQG decoder module $\mathcal{G}^{\text{dec}}(\cdot)$. Afterward, the DQG decoder module $\mathcal{G}^{\text{dec}}(\cdot)$ calculates the likelihoods of $\text{dic}_c$ for $t$-th word ($t = 1, \dots, T^{\mathcal{G}}$; $T^{\mathcal{G}}$ being the number of estimated words) $l_t \in \mathcal{R}^C$ as follows: $l_t = \mathcal{G}^{\text{dec}}(f^{\mathcal{G}})$. Next, we screen the initial retrieval results by comparing the automatically annotated answer labels in each candidate image and feedback answers from the user for the estimated question. The answer labels $a_n$ can be obtained via the QA modules $\{\mathcal{A}_{\text{word}}(\cdot), \mathcal{A}_{\text{dec}}(\cdot)\}$. First, we convert $l_t$ to one-hot vector $h_t \in \mathcal{R}^C$ for propagating the estimated question to the QA modules. In the test phase, for determining the $t$-th word, we convert $l_t$ to one-hot vector $h_t$ via the $\arg \max(\cdot)$ function. Here, since the $\arg \max(\cdot)$

function is a nondifferentiable procedure, it cannot be used in the training phase. Therefore, in the training phase, we convert $l_t$ to one-hot vector $h_t$ via differentiable Gummbel-Softmax-based hard sampling [1] [19]. We obtain the answer labels $a_n$ from $h_t$ and the candidate images $I_n$ as follows:

$$a_n = \mathcal{A}_{\text{dec}}(F^{\mathcal{A}}, I_n), \qquad (6)$$

$$F^{\mathcal{A}} = [f_1^{\mathcal{A}}, \dots, f_T^{\mathcal{A}}], \qquad (7)$$

$$f_t^{\mathcal{A}} = h_t F^{\text{word}}, \qquad (8)$$

where $F^{\text{word}}$ converts input words into the $D^{\text{word}}$-dimensional features (described in Subsec. 3.1). Similarly, we obtain the feedback answer $a^{\text{user}}$ targeted toward the inferred question. In the test phase, by extracting the maximum index of the estimated likelihoods $l_t$ via the $\arg \max_t(\cdot)$ function, the proposed approach obtains the generated question $q_t$. The generated question $q_t$ is then presented to the user, and the user gives the feedback answer $a^{\text{user}}$ targeted toward the question. In contrast, in the training phase, it takes a lot of time to manually prepare answers until the training convergence. By considering the QA module as users, we denote the answer label $a_n$ corresponding to the target image as $a^{\text{user}}$.

Finally, we calculate the screening similarities $\hat{s}_n$ based on the initial similarities $s_n$ as follows:

$$\hat{s}_n = s_n + \beta \text{sim}^{\text{text}}(a^{\text{user}}, a_n). \qquad (9)$$

where $\beta$ is a hyperparameter that balances the effect of the screening, and $\text{sim}^{\text{text}}(\cdot)$ is a text similarity function. We rank the candidate images $I_n$ as $\hat{R}_k$ in descending order of $\hat{s}_n$.

## 3.3 Optimization strategy

To generate questions that can screen similar images in DB, we train the DQG decoder module $\mathcal{G}^{\text{dec}}(\cdot)$ using pseudo image-query texts pairs prepared for the target DB. Since query text labels are generally not contained in each target DB, we prepare pseudo query text labels based on the candidate images in the target DB. Specifically, we generate texts $T_n^{\text{psu}}$ that represent candidate images

---
[1]https://pytorch.org/docs/stable/generated/torch.nn.functional.gumbel_softmax.html

$I_n$ via the pre-trained image description module I2T($\cdot$). Namely, $T_n^{psu}$ and $I_n$ are pseudo pairs that can be automatically prepared from the target DB.

Based on the calculated similarities $\hat{s}_n$ and the inferred one-hot vector $\boldsymbol{h}_t$, we calculate a loss $L$, consisting of two types of loss: retrieval loss $L^{rank}$ and gramatical loss $L^{\mathcal{D}}$, as follows:

$$L = \gamma L^{rank} + (1 - \gamma) L^{\mathcal{D}}, \qquad (10)$$

where $\gamma$ is a hyperparameter that balances the importance of loss.

Following the conventional text-based image retrieval methods, for all pseudo query text labels $T_n^{psu}$, we calculate the retrieval loss $L^{rank}$ as follows:

$$L^{rank} = \sum_m \begin{cases} \max\{0, \delta - \hat{s}_n + \hat{s}_m\} & (n \neq m) \\ 0 & (n = m) \end{cases}, \qquad (11)$$

where $\delta$ is a margin hyperparameter and $m = 1, 2, \ldots, N$. By training the DQG decoder module $\mathcal{G}^{dec}(\cdot)$ to minimize the re-ranking loss $L^{rank}$, $\mathcal{G}^{dec}(\cdot)$ can generate questions that can screen similar but non-target images.

Although the retrieval loss $L^{rank}$ helps with the improvement of the retrieval performance, there is no guarantee that the generated questions are grammatically correct for users. Therefore, DQG modules can generate questions that are incorrect for users using only the $L^{rank}$. To avoid such occurrence, we introduce grammatical loss $L^{\mathcal{D}}$ that can guarantee the grammatical correctness of the generated questions. Specifically, for all generated questions corresponding to $T_n^{psu}$, we calculate the grammatical loss $L^{\mathcal{D}}$ using the trained discriminative module $\hat{\mathcal{D}}(\cdot)$ (described in Subsec. 3.1) as follows:

$$L^{\mathcal{D}} = -\log \hat{\mathcal{D}}(q_t). \qquad (12)$$

Here, the discriminative module $\hat{\mathcal{D}}(\cdot)$ is trained to output a higher and lower value for the real and non-real questions, respectively. Namely, if the generated questions are similar to the real questions, $L^{\mathcal{D}}$ becomes lower. By introducing $L^{\mathcal{D}}$, it is expected that the generated questions are grammatically correct for users.

## 4 EXPERIMENTS

To evaluate the effectiveness of our DQG-based exact image retrieval, we conducted experiments using two major open datasets and one dataset with similar images. Specifically, we defined and evaluated the following research questions.

(1) Whether our approach can enhance the performance of the exact text-based image retrieval methods? (Subsec. 4.2)
(2) Whether the questions generated by our approach are grammatically correct or not? (Subsec. 4.3)
(3) Whether our approach is effective for DBs with a lot of similar images? (Subsec. 4.4)
(4) Whether our approach can effectively screen similar images than the conventional re-ranking approaches? (Subsec. 4.5)

### 4.1 Experimental settings

**Dataset.** Following the recent text-based image retrieval methods, we used the following two large-scale datasets with text-image pairs: MSCOCO and Visual Genome.

**Table 1: Experimental results for R@$k$, mean rank, and median rank using MSCOCO dataset.**

| | R@1 | R@10 | Mean | Median |
|---|---|---|---|---|
| PVSE [38] | 0.324 | 0.759 | 20.463 | 3 |
| PVSE+Ours w/o opt | 0.393 | 0.809 | 15.195 | 2 |
| PVSE+Ours w/o $L^{\mathcal{D}}$ | 0.479 | 0.860 | 9.771 | **1** |
| **PVSE+Ours** | **0.481** | **0.864** | **9.515** | **1** |
| SAN [20] | 0.337 | 0.777 | 23.143 | 2 |
| SAN+Ours w/o opt | 0.397 | 0.814 | 14.293 | 2 |
| SAN+Ours w/o $L^{\mathcal{D}}$ | 0.452 | 0.836 | 12.663 | 2 |
| **SAN+Ours** | **0.499** | **0.865** | **9.193** | **1** |
| VSRN [25] | 0.403 | 0.701 | 15.323 | **1** |
| VSRN+Ours w/o opt | 0.455 | 0.825 | 9.995 | **1** |
| VSRN+Ours w/o $L^{\mathcal{D}}$ | 0.503 | 0.854 | 7.524 | **1** |
| **VSRN+Ours** | **0.521** | **0.868** | **6.777** | **1** |
| PCME [9] | 0.379 | 0.735 | 20.632 | 3 |
| PCME+Ours w/o opt | 0.389 | 0.800 | 14.158 | 2 |
| PCME+Ours w/o $L^{\mathcal{D}}$ | 0.427 | 0.837 | 11.223 | 2 |
| **PCME+Ours** | **0.475** | **0.858** | **8.339** | **1** |
| SGM [8] | 0.352 | 0.765 | 25.322 | 3 |
| SGM+Ours w/o opt | 0.376 | 0.786 | 20.421 | 2 |
| SGM+Ours w/o $L^{\mathcal{D}}$ | 0.413 | 0.823 | 13.422 | 2 |
| **SGM+Ours** | **0.441** | **0.851** | **9.551** | **1** |
| DiVE [21] | 0.412 | 0.720 | 17.431 | 2 |
| DiVE+Ours w/o opt | 0.424 | 0.838 | 9.312 | **1** |
| DiVE+Ours w/o $L^{\mathcal{D}}$ | 0.476 | 0.852 | 8.395 | **1** |
| **DiVE+Ours** | **0.510** | **0.875** | **6.983** | **1** |
| CLIP [33] | 0.378 | 0.722 | 26.421 | 3 |
| CLIP+Ours w/o opt | 0.392 | 0.764 | 21.011 | 2 |
| CLIP+Ours w/o $L^{\mathcal{D}}$ | 0.402 | 0.814 | 14.502 | 2 |
| **CLIP+Ours** | **0.436** | **0.846** | **9.744** | **1** |
| BLIP [24] | 0.402 | 0.753 | 18.773 | 2 |
| BLIP+Ours w/o opt | 0.423 | 0.792 | 16.532 | 2 |
| BLIP+Ours w/o $L^{\mathcal{D}}$ | 0.498 | 0.841 | 12.331 | 1 |
| **BLIP+Ours** | **0.531** | **0.867** | **8.631** | **1** |

**MSCOCO [28]**

The MSCOCO dataset consists of images and corresponding texts that describe the contents of a paired image. This dataset is adopted by the most recent text-based image retrieval methods. Following the widely used data splits provided by [22], 123,287 and 5,000 images are used for the pre-training and target DB, respectively.

**Visual Genome [23]**

The Visual Genome dataset consists of images and corresponding texts that describe the particular region of the paired image. Following the conventional methods, 75,578 and 32,422 images are respectively used for the target DB [48, 57]. Note that since Visual Genome dataset does not have question and answer labels, we utilized the MSCOCO pre-training dataset for pre-training each module.

**Implementation details.** We compared our approach with some recently proposed text-based image retrieval methods [8,

**Table 2: Experimental results for R@$k$, mean rank, and median rank using the Visual Genome dataset.**

|  | R@1 | R@10 | Mean | Median |
|---|---|---|---|---|
| PVSE [38] | 0.0280 | 0.129 | 2943.513 | 258 |
| PVSE+Ours w/o opt | 0.0431 | 0.179 | 2742.562 | 190 |
| PVSE+Ours w/o $L^{\mathcal{D}}$ | 0.0985 | 0.221 | 2504.593 | 176 |
| **PVSE+Ours** | **0.104** | **0.234** | **2485.331** | **163** |
| SAN [20] | 0.0286 | 0.130 | 2861.242 | 248 |
| SAN+Ours w/o opt | 0.0442 | 0.180 | 27694.322 | 185 |
| SAN+Ours w/o $L^{\mathcal{D}}$ | 0.114 | 0.225 | 2634.507 | 164 |
| **SAN+Ours** | **0.122** | **0.242** | **2433.492** | **153** |
| VSRN [25] | 0.0390 | 0.130 | 2861.242 | 248 |
| VSRN+Ours w/o opt | 0.0472 | 0.185 | 2800.293 | 176 |
| VSRN+Ours w/o $L^{\mathcal{D}}$ | 0.120 | 0.233 | 2421.063 | 152 |
| **VSRN+Ours** | **0.124** | **0.248** | **2394.391** | **147** |
| PCME [9] | 0.0304 | 0.142 | 2755.495 | 249 |
| PCME+Ours w/o opt | 0.0451 | 0.182 | 2694.291 | 182 |
| PCME+Ours w/o $L^{\mathcal{D}}$ | 0.0895 | 0.209 | 2554.322 | 167 |
| **PCME+Ours** | **0.115** | **0.236** | **2423.391** | **154** |
| SGM [8] | 0.0341 | 0.131 | 2822.341 | 253 |
| SGM+Ours w/o opt | 0.0432 | 0.162 | 2704.321 | 221 |
| SGM+Ours w/o $L^{\mathcal{D}}$ | 0.0823 | 0.201 | 2563.432 | 181 |
| **SGM+Ours** | **0.102** | **0.229** | **2499.32** | **164** |
| DiVE [21] | 0.0398 | 0.132 | 2845.554 | 250 |
| DiVE+Ours w/o opt | 0.0758 | 0.183 | 2685.439 | 178 |
| DiVE+Ours w/o $L^{\mathcal{D}}$ | 0.0832 | 0.210 | 2497.889 | 178 |
| **DiVE+Ours** | **0.144** | **0.238** | **2338.482** | **160** |
| CLIP [33] | 0.0405 | 0.151 | 2742.491 | 231 |
| CLIP+Ours w/o opt | 0.0574 | 0.176 | 2698.752 | 215 |
| CLIP+Ours w/o $L^{\mathcal{D}}$ | 0.0968 | 0.205 | 2477.231 | 175 |
| **CLIP+Ours** | **0.112** | **0.247** | **2313.441** | **146** |
| BLIP [24] | 0.0454 | 0.173 | 2652.311 | 227 |
| BLIP+Ours w/o opt | 0.0603 | 0.181 | 2534.123 | 211 |
| BLIP+Ours w/o $L^{\mathcal{D}}$ | 0.0994 | 0.212 | 2405.123 | 165 |
| **BLIP+Ours** | **0.121** | **0.264** | **2223.486** | **128** |

9, 20, 21, 24, 25, 33, 38]. Note that we implemented all comparative methods based on the open-source codes provided by the authors. Furthermore, to evaluate the effectiveness of our training, we compared the retrieval performance calculated using the DQG module without optimization in Subsecs. 3.2 and 3.3 (hereinafter referred to as Ours w/o opt). Namely, it conducts the re-ranking with the DQG module after the pre-training in Subsec. 3.1. Also, to evaluate the effectiveness of $L^{\mathcal{D}}$, our approach without $L^{\mathcal{D}}$ is used for comparison (hereinafter referred to as Ours w/o $L^{\mathcal{D}}$). In our approach, the DQG, QA, and image description modules are constructed following [26], [34], and [10], respectively. Also, we applied the transformer network for our discriminative model. For the parameters in the equations, we set $\gamma = 0.3$ following the conventional researches and experimentally set $\alpha = 0.9$, $\beta = 0.6$, and $\delta = 0.3$. In place of the function $\mathrm{sim}^{\mathrm{text}}(\cdot)$, we used a cosine similarity function using GloVe [31] word features. Each module is pre-trained following their papers, and our training is conducted until the losses converged.

**Answer preparation.** To evaluate the effectiveness of our approach, answers to the generated questions should be required.

However, preparing answers to all generated questions comes at a great cost. Even if we manually prepare the answers to all generated questions, the experimental objectivity and repeatability will not be guaranteed since such manual preparation is always required for considering novel approaches. To ensure the experimental objectivity and repeatability, inspired from the dialog system researches [14, 40], we prepared answers to the generated questions by assuming the QA model as a user. To avoid overfitting toward the QA module and to ensure experimental objectivity, we use the QA model that has different QA module architectures in the training phase ($\{\mathcal{A}^{\mathrm{word}}(\cdot), \mathcal{A}^{\mathrm{dec}}(\cdot)\}$). However, using the different model architectures in the training and evaluation, there is no guarantee that the QA model will always answer the presented questions accurately and that its model is closer to the user behavior. Even if we assume the QA model as a user, our approach cannot unfairly identify the target images and does not use any label information.

## 4.2 Comparison with conventional text-based image retrieval methods

We first evaluate the research question "Whether our approach can further enhance the text-based image retrieval performance ?" by introducing our approach to the conventional text-based image retrieval methods. Mean rank, median rank, and Recall@$k$(R@$k$) are used for the evaluation metrics, following the conventional text-based image retrieval methods. R@$k$ is defined as follows:

$$R@k = \frac{g_k}{J} \quad (k = 1, 2, \dots, N), \tag{13}$$

where $J$ and $g_k$ represent the number of queries for evaluation and the number of queries that can rank relevant images in the top-$k$ retrieval results, respectively. Here, we define a relevant image as an image associated with a query text in datasets. Namely, there is only one paired image in $N$ candidate images for each text query. Note that, since there is only one ground truth for each query, the other evaluation metrics (e.g., MAP, NDCG) are not used in the recent text-based image retrieval methods.

Experimental results with MSCOCO and Visual Genome datasets are shown in Tables 1 and 2, respectively. In each table, "Kiros '14 +Ours" reveals results using the method by Kiros '14 and our approach, respectively. We can see that our approach can drastically enhance the retrieval performance of all baseline text-based image retrieval methods as shown in Tables 1 and 2. Specifically, the enhancement of mean and median ranks reveals that our approach stably and effectively distinguishes similar contents in the target DB. Furthermore, we observe that our approach contributes to the improvement of the retrieval performance compared with "Ours w/o opt." These results mean that the optimization with our loss is effective for screening. Surprisingly, our approach outperforms "Ours w/o $L^{\mathcal{D}}$". Thus, introducing $L^{\mathcal{D}}$ is also effective for enhancing retrieval performance. From the results obtained after introducing $L^{\mathcal{D}}$, we consider that $L^{\mathcal{D}}$ has the regularization ability to prevent overfitting toward the QA modules. Further analysis of the results can lead to further improvement of the retrieval performance. Additionally, examples of retrieval results are also shown in Fig. 1. From these results, we confirmed that our approach can improve the retrieval performance of the conventional text-based image retrieval methods in various datasets.

**Table 3: Experimental results for DScore in MSCOCO and Visual Genome dataset. Since "Ours w/o opt" can be considered as upper limits of the other methods, we also show the margin between "Ours w/o opt" and the other methods.**

| | DScore in MSCOCO | DScore in Visual Genome |
|---|---|---|
| PVSE+Ours w/o opt | 0.901 | 0.948 |
| PVSE+Ours w/o $L^{\mathcal{D}}$ | 0.293 (-0.708) | 0.228 (-0.720) |
| **PVSE+Ours** | **0.899 (-0.002)** | **0.930 (-0.018)** |
| SAN+Ours w/o opt | 0.999 | 0.985 |
| SAN+Ours w/o $L^{\mathcal{D}}$ | 0.492 (-0.507) | 0.300 (-0.685) |
| **SAN+Ours** | **0.993 (-0.006)** | **0.984 (-0.001)** |
| VSRN+Ours w/o opt | 0.979 | 0.969 |
| VSRN+Ours w/o $L^{\mathcal{D}}$ | 0.429 (-0.550) | 0.325 (-0.644) |
| **VSRN+Ours** | **0.968 (-0.011)** | **0.960 (-0.009)** |
| PCME+Ours w/o opt | 0.992 | 0.990 |
| PCME+Ours w/o $L^{\mathcal{D}}$ | 0.293 (-0.699) | 0.392 (-0.598) |
| **PCME+Ours** | **0.989 (-0.003)** | **0.979 (-0.019)** |
| SGM+Ours w/o opt | 0.994 | 0.991 |
| SGM+Ours w/o $L^{\mathcal{D}}$ | 0.231 (-0.763) | 0.330 (-0.661) |
| **SGM+Ours** | **0.990 (-0.004)** | **0.983 (-0.008)** |
| DiVE+Ours w/o opt | 0.997 | 0.993 |
| DiVE+Ours w/o $L^{\mathcal{D}}$ | 0.294 (-0.703) | 0.276 (-0.717) |
| **DiVE+Ours** | **0.989 (-0.008)** | **0.987 (-0.006)** |
| CLIP+Ours w/o opt | 0.993 | 0.992 |
| CLIP+Ours w/o $L^{\mathcal{D}}$ | 0.342 (-0.651) | 0.363 (-0.629) |
| **CLIP+Ours** | **0.991 (-0.002)** | **0.987 (-0.005)** |
| BLIP+Ours w/o opt | 0.996 | 0.992 |
| BLIP+Ours w/o $L^{\mathcal{D}}$ | 0.203 (-0.793) | 0.342 (-0.650) |
| **BLIP+Ours** | **0.993 (-0.003)** | **0.981 (-0.011)** |

## 4.3 Evaluating the grammatically correctness of the generated questions

Next, we confirm the research question "Whether the questions generated by our approach are grammatically correct or not ?". To confirm them, following the automatic evaluation metrics in the field of natural language processing [6], we calculated the evaluation metrics: DScore, using a discriminative model $\hat{\mathcal{D}}^{\text{eva}}(\cdot)$. This model $\hat{\mathcal{D}}^{\text{eva}}(\cdot)$ is trained following the same procedure of $\hat{\mathcal{D}}(\cdot)$ in Subsec. 3.1. To avoid overfitting toward the discriminative module in the training phase $\hat{\mathcal{D}}(\cdot)$ and ensuring experimental objectivity, we trained $\hat{\mathcal{D}}^{\text{eva}}(\cdot)$ with the dataset different from $\hat{\mathcal{D}}(\cdot)$ and applied model architectures different from $\hat{\mathcal{D}}(\cdot)$. DScore is calculated as follows:

$$\text{DScore} = \frac{1}{J} \sum_j \hat{\mathcal{D}}^{\text{eva}}(\text{q}_{j,t}^{\text{eva}}) \quad (j = 1, 2, \ldots, J), \quad (14)$$

where $\text{q}_{j,t}^{\text{eva}}$ represents the question generated when the $j$-th query is inputted for our approach. Namely, the range of DScore is 0 to 1. Note that when we test $\hat{\mathcal{D}}^{\text{eva}}(\cdot)$ to determine whether $\hat{\mathcal{D}}^{\text{eva}}(\cdot)$ can distinguish between $10,000$ real questions $q^{\text{real}}$ and $10,000$ grammatically incorrect questions $q^{\text{gen}}$ (defined in Subsec. 3.1), $\hat{\mathcal{D}}^{\text{eva}}(\cdot)$ can accurately distinguish 99.9% of the samples. These results confirm that $\hat{\mathcal{D}}^{\text{eva}}(\cdot)$ has high distinguishing performance, and DScore has high reliability.

**Table 4: Experimental results for R@$k$, mean rank and median rank using the biased-MSCOCO DB.**

| | R@1 | R@10 | Mean | Median |
|---|---|---|---|---|
| PVSE [38] | 0.343 | 0.761 | 15.810 | 2 |
| PVSE+Ours w/o opt | 0.359 | 0.779 | 14.104 | 2 |
| PVSE+Ours w/o $L^{\mathcal{D}}$ | 0.438 | 0.844 | 12.204 | **1** |
| **PVSE+Ours** | **0.462** | **0.864** | **9.320** | **1** |
| SAN [20] | 0.347 | 0.768 | 15.110 | 2 |
| SAN+Ours w/o opt | 0.352 | 0.793 | 14.392 | 2 |
| SAN+Ours w/o $L^{\mathcal{D}}$ | 0.455 | 0.831 | 9.994 | **1** |
| **SAN+Ours** | **0.471** | **0.842** | **9.603** | **1** |
| VSRN [25] | 0.379 | 0.793 | 14.332 | 2 |
| VSRN+Ours w/o opt | 0.381 | 0.803 | 12.331 | 2 |
| VSRN+Ours w/o $L^{\mathcal{D}}$ | 0.499 | 0.877 | 7.506 | **1** |
| **VSRN+Ours** | **0.504** | **0.885** | **7.417** | **1** |
| PCME [9] | 0.349 | 0.751 | 17.422 | 3 |
| PCME+Ours w/o opt | 0.363 | 0.779 | 15.445 | 2 |
| PCME+Ours w/o $L^{\mathcal{D}}$ | 0.431 | 0.796 | 11.445 | 2 |
| **PCME+Ours** | **0.469** | **0.861** | **10.338** | **1** |
| SGM [8] | 0.348 | 0.749 | 18.551 | 3 |
| SGM+Ours w/o opt | 0.358 | 0.761 | 16.532 | 2 |
| SGM+Ours w/o $L^{\mathcal{D}}$ | 0.423 | 0.778 | 12.421 | 2 |
| **SGM+Ours** | **0.464** | **0.853** | **11.002** | **1** |
| DiVE [21] | 0.380 | 0.790 | 15.684 | 2 |
| DiVE+Ours w/o opt | 0.387 | 0.795 | 14.009 | 2 |
| DiVE+Ours w/o $L^{\mathcal{D}}$ | 0.434 | 0.833 | 11.041 | 2 |
| **DiVE+Ours** | **0.458** | **0.849** | **9.381** | **1** |
| CLIP [33] | 0.354 | 0.752 | 17.531 | 3 |
| CLIP+Ours w/o opt | 0.361 | 0.772 | 16.521 | 2 |
| CLIP+Ours w/o $L^{\mathcal{D}}$ | 0.411 | 0.813 | 11.313 | **1** |
| **CLIP+Ours** | **0.474** | **0.834** | **10.021** | **1** |
| BLIP [24] | 0.385 | 0.763 | 14.212 | 2 |
| BLIP+Ours w/o opt | 0.394 | 0.785 | 13.412 | 2 |
| BLIP+Ours w/o $L^{\mathcal{D}}$ | 0.432 | 0.842 | 9.331 | **1** |
| **BLIP+Ours** | **0.489** | **0.874** | **8.411** | **1** |

Experimental results of the DScore using MSCOCO and Visual Genome datasets are shown in Table 3. Note that since "Ours w/o opt" uses a model that is pre-trained for generating questions similar to the actual questions, and the other methods are trained based on "Ours w/o opt", "Ours w/o opt" can be considered as upper limits of the other methods. Therefore, we also shows the margin between "Ours w/o opt" and the other methods in Table 3. As shown in Table 3, "Ours w/o $L^{\mathcal{D}}$" significantly underperforms "Ours w/o opt" in all text-based image retrieval methods. This means that although the optimization only with $L^{\text{rank}}$ enhances the retrieval performance of the baseline text-based image retrieval methods, it results in ignoring the linguistic reasonability. Besides, "Ours" outperforms "Ours w/o $L^{\mathcal{D}}$" and reaches similar performance of "Ours w/o opt." These results reveal that the introduction of $L^{\mathcal{D}}$ is effective for assuring grammatical correctness. Considering the fact that $L^{\mathcal{D}}$ can also improve the retrieval performance, we can say that $L^{\mathcal{D}}$ is a significant loss in our approach. From these results, we confirmed that the generated questions by our approach are grammatically correct.

## 4.4 Evaluating the effectiveness towards DBs with similar contents

Evaluations of our approach are conducted based on two large-scale datasets in Subsec. 4.2. Although the effectiveness of our approach can be verified for these large-scale datasets, whether our approach is effective for DBs with a lot of similar contents (hereinafter referred to as biased-DBs) is not guaranteed. In this subsection, we verify them by conducting experiments on the biased-DBs. To the best of our knowledge, publicly available biased-DBs do not exist. Therefore, we construct an artificially biased-DB by assuming DBs with similar objects are biased-DBs.

For constructing the artificially biased-DB, an object label contained in most images of the MSCOCO target DB is calculated, and we reconstructed the MSCOCO target DB so that the images of its DB absolutely contain the object label. The calculated label is "person", and the number of the extracted images is 2, 628. We define these extracted images as a biased-MSCOCO DB. Namely, the images in the biased-MSCOCO DB include similar contents related to "person". We compare the retrieval performance using the biased-MSCOCO DB. Also, the other settings follow Subsec. 4.2.

The experimental results are shown in Table 4. As shown in this table, the same trend as in the two large-scale datasets can be observed in the biased-MSCOCO DB. From these results, we have verified the effectiveness of the proposed approach for a biased-DB.

## 4.5 Comparison with feedback-based re-ranking approaches

For evaluating the research question "Whether our approach can effectively screen similar but non-target images than the conventional re-ranking approaches ?", we compare our approach with the conventional feedback-based re-ranking approaches. Broadly, comparing the re-ranking approaches that receive different feedback is extremely difficult [35]. Therefore, the experiments were conducted so as to maximize the performance of each comparative method [15, 27, 29, 32, 47, 50, 52] as far as possible. Most conventional approaches re-rank the retrieval results by asking users to select relevant images in the top-ranked images. For realizing the above procedure, in our experiments, we considered images containing the same object labels of the target image as relevant images following the experiments in [50, 52]. In the experiments, at first, the initial retrieval results are computed based on the text-based image retrieval method of PVSE [38]. Next, the initial retrieval results were re-ranked based on each re-ranking approach.

Tables 5, 6 and 7 show experimental results on MSCOCO, Visual Genome, and biased-MSCOCO DB. As shown in each table, we can see that our approach improves the retrieval performance of baseline methods compared with the conventional approaches. These results imply that our approach is more effective for enhancing the retrieval performance of the conventional text-based image retrieval methods than the conventional re-ranking approaches.

## 5 CONCLUSIONS

In this paper, we introduced a novel approach called Database Question Generation (DQG) to enhance the performance of exact text-based image retrieval systems. Our approach learn the appropriate question on the target DB only from those candidate images,

**Table 5: Comparison with feedback-based re-ranking approaches using the MSCOCO dataset.**

|  | R@1 | R@10 | Mean | Median |
|---|---|---|---|---|
| Initial result | 0.317 | 0.759 | 20.463 | 2 |
| NN + BQS [15] | 0.316 | 0.655 | 20.052 | 2 |
| SVM [27] | 0.275 | 0.605 | 39.701 | 3 |
| EMR [47] | 0.317 | 0.765 | 20.534 | 2 |
| PRF [29] | 0.315 | 0.650 | 19.110 | 2 |
| RFNet [32] | 0.278 | 0.614 | 35.435 | 3 |
| DBQA [50] | 0.389 | 0.851 | 13.702 | **1** |
| RQA [52] | 0.412 | 0.858 | 12.221 | **1** |
| **Ours** | **0.481** | **0.864** | **9.515** | **1** |

**Table 6: Comparison with feedback-based re-ranking approaches using the Visual Genome dataset.**

|  | R@1 | R@10 | Mean | Median |
|---|---|---|---|---|
| Initial result | 0.0280 | 0.129 | 2943.513 | 258 |
| NN + BQS [15] | 0.0283 | 0.129 | 2852.621 | 255 |
| SVM [27] | 0.0205 | 0.0952 | 4109.735 | 610 |
| EMR [47] | 0.0287 | 0.132 | 2899.920 | 245 |
| PRF [29] | 0.0282 | 0.130 | 2892.026 | 247 |
| RFNet [32] | 0.0208 | 0.0966 | 4001.001 | 548 |
| DBQA [50] | 0.0327 | 0.173 | 2846.555 | 234 |
| RQA [52] | 0.0654 | 0.201 | 2679.44 | 201 |
| **Ours** | **0.104** | **0.234** | **2485.331** | **163** |

**Table 7: Comparison with feedback-based re-ranking approaches using the biased-MSCOCO DB.**

|  | R@1 | R@10 | Mean | Median |
|---|---|---|---|---|
| Initial result | 0.343 | 0.768 | 15.810 | 2 |
| NN + BQS [15] | 0.334 | 0.652 | 16.867 | 2 |
| SVM [27] | 0.0980 | 0.278 | 169.613 | 28 |
| EMR [47] | 0.341 | 0.653 | 15.464 | 2 |
| PRF [29] | 0.336 | 0.654 | 16.326 | 2 |
| RFNet [32] | 0.104 | 0.341 | 141.485 | 27 |
| DBQA [50] | 0.394 | 0.801 | 12.770 | 2 |
| RQA [52] | 0.437 | 0.833 | 11.422 | 2 |
| **Ours** | **0.462** | **0.864** | **9.320** | **1** |

facilitating the screening of similar but non-target images. By responding to these generated questions, users can effectively retrieve their desired images, even from queries with limited information. Our experimental results underscore the efficacy of our approach in significantly improving exact text-based image retrieval performance across diverse datasets, confirming its potential in advancing the state-of-the-art in image retrieval methodologies. While our study marks a crucial step forward, our current model and loss architectures represent initial implementations and can benefit from refinement and sophistication. Additionally, in-depth analyses and discussions on the grammatical correctness and semantic coherence of the generated questions would contribute significantly to the robustness of our approach. For future works, we will tackle both architecture improvement and further grammatical correctness verification.

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
