# OpenReview forum: "DQG: Database Question Generation for Exact Text-based Image Retrieval"
_acmmm.org/ACMMM/2024/Conference — MM2024 Oral_

### Official Review · Reviewer_xeqy · 2024-05-14

**Rating:** 5
**Confidence:** 2

**Summary:**

In this paper, the authors introduced a novel approach called Database Question Generation (DQG) to enhance the performance of exact text-based image retrieval systems. The proposed approach learns the appropriate question on the target DB only from those candidate images facilitating the screening of similar but non-target images. The experimental results demonstrate the efficacy of the approach in significantly improving exact text-based image retrieval performance across diverse datasets.

**Strengths:**

The main strengths of this paper include,
1. the proposed DQG approach for exact text-based image retrieval was developed.
2. the new DQG learning scheme without task-specific labels
3. significant performance improvement

**Limitations:**

Here are some limitations in the text:

1. Lack of Comparative Analysis with Recent Advances: while the paper compares the DQG approach with a few existing methods, it lacks a thorough comparative analysis with the latest state-of-the-art models that utilize newer techniques in recent years for the same task. only DiVE was published in 2023, most of the rest methods were developed in 2019 and 2021.

2. Potential Bias and Overfitting: The method's reliance on pre-trained modules and the absence of a robust validation for the discriminative model might lead to overfitting or biases in question generation. The authors of this paper do not sufficiently discuss measures to mitigate these risks.

**Suitability:**

3

---

### Official Review · Reviewer_kFxg · 2024-05-24

**Rating:** 4
**Confidence:** 2

**Summary:**

This paper introduces a DQG learning scheme without task-specific labels and DQG question generation model, which reach desired images by answering generated questions, further to enhance exact cross-modal image retrieval performance. Experimental results on publicly available datasets show that DOG model improve image retrieval performance. The motivation is relatively novel, and experiments result shows the proposed method is effective.

**Strengths:**

1. This paper introduces a learning scheme without task-specific labels, which to generate the suitable questions from DB information input.
2. This paper proposes a DQG model to enhance exact cross-modal image retrieval performance, which reach desired images by answering generated questions.
3. The proposed model do experiments on different dataset and compared result with popular methods.

**Limitations:**

1. How to ensure the quality of the generated question, and is it taken into account when building the dataset?
2. No clear explanation with how to generate suitable questions without any question labels for retrieval tasks.

**Suitability:**

2

---

### Official Review · Reviewer_vq5W · 2024-05-24

**Rating:** 3
**Confidence:** 3

**Summary:**

This paper discusses a significant challenge in Text-based Image Retrieval (TBIR) systems: how to enhance the system's performance during precise image retrieval tasks. To address this issue, the paper proposes a novel solution, namely integrating a Question Generation (QG) module. This module generates questions that help the system better understand the user's retrieval intentions, thereby improving the accuracy of the retrieval.

**Strengths:**

1. **Enhanced User Interaction**: The idea of generating targeted questions to guide users in providing more specific information is intriguing. It has proven effective in the domain of document information retrieval but is relatively rare in image retrieval. This method could also increase user interaction with the system, thus enhancing retrieval precision.

2. **Extensive experimental results and supplementary materials**: Comprehensive experiments and provided materials make this work more convincing.

**Limitations:**

1. **User Burden**: Although the Question Generation module can help improve retrieval accuracy, it may also increase the user's operational burden. Users need to respond to questions generated by the system, which could lead to a decline in user experience, especially in scenarios where quick retrieval results are required.

2. **User Motivation**: Users have two options: 1) browse N photos and then make a decision (do not use DQG), or 2) answer questions generated by the system (use DQG). The authors should explain the advantages of option 2 over option 1, thereby supporting the motivation for this paper.

3. **Issues with Article Writing**: The methods described in this paper (beyond the initial text-based image retrieval) involve complex structures and processes, such as the DQG-encoder, DQG-decoder, QA-module, and Discriminative module. The inputs and outputs of each module, the insights behind their designs, and specific technical details should be clearly explained. The paper is very brief in introducing these modules, merely describing the pipeline process, making it difficult to understand the design and principles of the methods.

4. **Other Questions**: 1) The initial text-based image retrieval generates N images; how is N determined? 2) The paper mentions the use of a transformer in the discriminative module, but what are the network structures of the other parts? 3) The main role of the DQG decoder is to generate questions; does this module use pre-trained model parameters or is it trained from scratch? 4) Why is grammaticality used to measure the quality of the generated questions in natural language aspects? To make the question easy to understand, grammar is just one aspect; overall fluency is equally important. Have the authors considered using NLP's perplexity metric, which might be more appropriate?

Overall, the ideas presented in this work are intriguing and worth exploring. They might have more interesting applications on mobile devices, such as for precise local image data retrieval, but there are shortcomings in the writing, motivation for use, and other issues mentioned above. This work is interesting and if the concerns above are addressed, I would be glad to raise the score.

**Suitability:**

3

---

### Official Review · Reviewer_FbEk · 2024-05-24

**Rating:** 3
**Confidence:** 4

**Summary:**

This paper introduces a DB question generation (DQG) model to enhance exact cross-modal image retrieval performance, which learns to generate effective questions that precisely screen similar but non-target images using DB contents information.

**Strengths:**

1. Motivation and problem formulation is solid, with examples that help readers understand.
2. The theoretical approach is reasonable.

**Limitations:**

1.The methods cited in the ablation experiments are a bit old. And in the comparison experiments in Table 5, except for RQA, other methods are relatively old，it is not very convincing.
2.The innovation is not obvious, and there is no detail on how to effectively control question quality.

**Suitability:**

3

---

### Meta-Review · Area_Chair_wBSD · 2024-07-03

**Recommendation:** Accept (Oral)
**Confidence:** 5

**Metareview:**

The authors seem to have done a good job addressing reviewers' concerns during the rebuttal phase, as reflected by a rating increase from the majority of the reviewers:

R1: from borderline reject to weak accept
R2: from borderline reject to weak accept
R3: from borderline accept to weak accept
R4: kept the original weak accept rating

However, for the fairness to other authors, the initial major concerns of the reviewers indicate the lack of clarify on the justification of the contributions and/or lack of technical clarity.

To summarize, this paper introduces a novel Deep Question Generation (DQG) learning scheme for text-based image retrieval without requiring task-specific labels, demonstrating performance improvements. While the approach shows promise, there are also a few limitations as pointed out by reviewers: 1) insufficient comparative study with SOTA methods in the initial submission, despite the additional experimental results presented during rebuttal, 2) concern about potential overfitting risks (would be nicer to theoretically provide insights into this, or provide cross-dataset validation results.)